# Re-examining the quality dimensions of synthetic speech

*Fritz Seebauer[1], Michael Kuhlmann[2], Reinhold Haeb-Umbach[2], Petra Wagner[1]*

[1]Bielefeld University, Germany
[2]Paderborn University, Germany

`{fritz.seebauer, petra.wagner}@uni-bielefeld.de, {kuhlmann,haeb}@nt.upb.de`

## Abstract

The aim of this paper is to generate a more comprehensive framework for evaluating synthetic speech. To this end, a line of tests resulting in an exploratory factor analysis (EFA) have been carried out. The proposed dimensions that encapsulate the construct of "synthetic speech quality" are: "human-likeness", "audio quality", "negative emotion", "dominance", "positive emotion", "calmness", "seniority" and "gender", with item-to-total correlations pointing towards "gender" being an orthogonal construct. A subsequent analysis on common acoustic features, found in forensic and phonetic literature, reveals very weak correlations with the proposed scales. Inter-rater and inter-item agreement measures additionally reveal low consistency within the scales. We also make the case that there is a need for a more fine grained approach when investigating the quality of synthetic speech systems, and propose a method that attempts to capture individual quality dimensions in the time domain.

**Index Terms**: speech synthesis evaluation, factor analysis, speech quality

## 1. Introduction

The evaluation of quality for any given speech synthesis system is commonly carried out on three dimensions. Its perceived naturalness, its quality and its intelligibility. With the advancements in text-to-speech (TTS) systems over the past decades, the problem of intelligibility has become almost redundant, with a main focus of research now lying on generating more natural sounding voices [1]. These advancements however are generally reported on old scales based on the ITU P.85 [2] for measuring signal degradation. These original scales and variations of it have become the standard for many challenges which offer a framework to compare state-of-the-art (SOTA) TTS systems [3, 4]. This is despite early criticism regarding the completeness and nature of these quality scales such as [5, 6, 7, 8]. Further efforts to rework the set of quality evaluation instruments have been carried out, but they are few and between [9, 10, 11] or date back to a time of diphone synthesis systems, which displayed their own specific set of degradations and might as such not been applicable to modern day systems [12, 13]. We take these critiques to warrant a re-examination of commonly used mean opinion scores (MOS) on modern day speech synthesis systems. Secondly, it has also been noted, that the scope of a critical speech unit (CSU) very much co-determines the outcome of a quality evaluation [14]. This need for a more context-aware and time-sensitive method of evaluation has also been discussed in [15] and the importance of the specific wording in a synthetic speech evaluation has been pointed out in [16]. We make the case for a change in evaluation procedure when constructing new systems, to gain a more fine grained understanding of what the actual shortcomings of the systems under evaluation are. This runs counter to the current practice which seems to have adapted a methodology of trying to maximise the MOS of a given system over previous iterations, without analysing why the changes occur. To address these shortcomings we propose a different evaluation technique which takes into account the temporal aspect of speech data, by having participants mark faulty segments for the previously determined quality dimensions, similar to [17]. This type of rating scheme would promise to offer further insight into the relationship between the perceptual quality dimensions of participants on the one side and acoustic or other signal related properties on the other.

## 2. Data

The employed corpus was comprised of 14 different TTS systems with varying accents, vocoders and training datasets. An overview can be surveyed in Table 1. We tried to ensure that a variety of modern day architectures are represented in the data set. Since the voice of a TTS system strongly depends on the underlying data, we also tried to cover a variety of commonly used data sets for TTS construction, omitting the older blizzard data sets for sparsity reasons. We also tried to incorporate multiple varieties of English. This was done to ensure that the resulting work of our experiments will not be purely based on US American voices, continuing a tradition of 'white washing' datasets that has plagued Machine Learning research for decades [18, 19]. The chosen content consisted of three Harvard sentences chosen from a set of 60 in total [20]. Each triple was separated by 500ms of silence in between each sentence. All systems generated 14 samples of these triple sets and the resulting signals were downsampled to 22050Hz and amplitude normalized to -18dB. We are unable to release a copy of the data set due to licence restrictions but have included all relevant details to enable comparison to similar data sets and tasks.

## 3. Methods

The analyses presented in this paper are twofold. First we conducted a series of experiments to obtain terms of quality for synthetic speech experiments in a bottom up fashion. These are subsequently examined to determine overarching perceptual dimensions of quality using exploratory factor analysis. Secondly we present a framework for capturing subjects impressions of these terms of quality in the time domain.

Table 1: *TTS system architectures.*

|     | Identifier | Vocoder | Dataset | Accent | Gender |
|-----|------------|---------|---------|--------|--------|
| 2x  | Google wavenet | unknown | unknown | GB+AU | M+M |
|     | Amazon Polly | unknown | unknown | ZA+IN | F+F |
|     | Microsoft Deepspeech | unknown | unknown | NZ+IR | F+M |
|     | Silero TTS | unknown | unknown | US+US | F+M |
| 1x  | vits | end-to-end | vctk | GB | M |
|     | fastspeech2 | end-to-end | LibriTTS | US | M |
|     | yourtts-multi | end-to-end | vctk | US | F |
|     | overflow | HifiGAN | LJ | US | F |
|     | speedy-speech | HifiGAN | LJ | US | F |
|     | espnet-xvector-transformer | MultibandMelGAN | LibriTTS | US | M |

## 3.1. Scale derivation

For deriving the original scale items we roughly followed the recommended procedure for inductive item generation outlined in [21], [22] and [23]. The original terms of quality yield from a pre-experiment in which 40 participants were asked to freely supply terms which they feel best encapsulate the quality of a given synthetic sample [24]. They were given a digital text input in which to denote the terms and instructed to supply at least three items per audio. The terms could be nouns, adjectives or even whole phrases. The items were converted into unilateral scales following the suggestions in [25], regarding the fact that participants tend to have an easier time identifying the existence or absence of a feature rather than giving an opinion. Polysemous items were further appended with a qualifier to ensure that participants would actually rate the same perceived trait, e.g. *funny/humorous* to avoid funny being interpreted as strange. These terms were then reduced with the employ of a 2h focus group interview of relevant experts. The panel was constructed of 1 speech technologist, 2 phoneticians and 2 clinical linguists. The group first discussed multiple contenders for a valid definition of "synthetic speech quality" [25, 26, 27, 28], to generate a shared base of discussion. The panel then surveyed an original set of 68 scales, comprised of the experiments items as well additional terms found in literature [25, 29], and examined them for relevancy and expected clarity to naïve participants reducing them down to 64. In a final pre-test these items were presented to 12 participants (5 experts, 7 naïve, L1=Mixed), with randomized synthetic audio samples from the corpus described in sec. 2. The participants were instructed to rate their impressions of the audio on the provided scales. They were then asked to denote the clarity of the scales' labels on a 4 point Likert scale from "very clear" to "very vague". The initial rating task was administered to help participants gather experience on the difficulties encountered while applying the scales to an audio task, instead of purely querying the semantic clarity. The resulting scores were averaged across participants by treating the ordinal levels as interval data. Computing the per scale certainty score with a previously determined dropout threshold of 25% of the maximum, no items were omitted, with the lowest scores falling to "dark" and "bright" at 71% each.

## 3.2. Factor Analysis

In order to group the scales into overarching dimensions of quality, 63 participants (32 M, 31F, L1=English) were recruited over the online platform prolific to rate 4 samples each, totaling 252 samples, or 18 per system. Given that the corpus only consisted of 15 distinct samples of each architecture, three ran-

dom audios per system obtained duplicate ratings. The slight skew in stratification balance regarding the spoken content was deemed insignificant and all ratings were used for further evaluation, yielding a rating to item ratio of 4:1. The scales were presented as continuous intervals using sliders, as there were no labels available to mark individual anchors within the scale, which would be necessary for a Likert scale and we assume the underlying dimensions to be continuous. Additionally, previous research on voice quality perception has found that participants tend to agree more on continuous scales [30]. The factor analysis was computed with oblique rotation, as we do not expect the factors to be orthogonal. To ensure the scales were actually correlated, a Bartletts test of sphericity was carried out which showed a high correlation with $p < 0.01$. Prior scree plot examination determined 8 factors to be the threshold from which the explained variance does not significantly increase. This hypothesis was confirmed using parallel analysis. During the experiment one randomly selected scale was duplicated for each sample to serve as a control according to the suggestions in [31]. Each participant's inconsistency score was computed by way of:

$$c = \sum_{i=1}^{n} \left( \frac{|x_i - x_{dup}|}{\sigma_i^2} \right) \qquad (1)$$

where $x_i$ and $x_{dup}$ describe the values for the original and duplicated scale, $\sigma_i^2$ the variance for that scale across participants and $n$ the total number of audios rated by that participant. As all participants rated the same amount of samples $n$ is constant across subjects. All participants whose inconsistency amounted to more than one fourth of the maximum possible divergence for the duplicated scales were excluded from the final analysis, removing four participants. The scale order was shuffled between items and participants to avoid context effects. Each participant was also presented with a training phase containing the same two anchor samples, which were not part of the analysis, to establish a consistent frame of reference for their responses similar to the suggestions in [30].

### 3.2.1. Consistency

To assess the consistency of the proposed scales across systems, we report the Intra Class Correlation (ICC) coefficient for the whole original set. This should give a measure of how invariant the dimensions are to the spoken content, as there were 15 distinct samples for each system in the original data set. To compute inter-rater consistency for the scales, a separate within subject design was employed. The systems were reduced to 4, due to the time constraints on a single participant. To retain as much variability as possible the content was randomly

selected between systems but kept same between participants. 20 (10m/10f, L1=Eng) naïve listeners were presented with the samples in a latin square design. Again, duplicated scales were introduced as control, removing 3 participants from the final analysis. The agreement results were evaluated by computing ICC(3,C) for the whole set, as well Krippendorff's alpha [32] for the individual scale items. The overall ICC denotes the overall agreement of the 20 participants across all scales, while the $\alpha$ coefficients are computed per scale across systems, depicting inter-rater agreement on interval scales, to obtain an estimate of which quality items would give consistent ratings in real experiment conditions.

### 3.2.2. Acoustic analysis

To gain some preliminary insight into how the perceptual dimensions interact with known acoustic measures, a correlation analysis on the derived quality scales was conducted. Five different acoustic measures were chosen, which are known to have strong explanatory power for small segments in the fields of speaker forensics and voice quality research [33, 34, 35] . The chosen measures consisted of periodicity markers: jitter, shimmer and spectral flux, spectral slope measurements in different frequency bands, cepstral peak prominence (CPP), as well as the fundamental frequency. Since the perceptual quality ratings always pertain to a whole file, the acoustic correlates were averaged over the whole duration, with spectral slope only being computed on voiced segments and spectral flux on voiced and unvoiced parts separately. Most features were computed using the openSMILE [36] python API. CPP was derived using the definitions in [37]. For the acoustic analysis, the Spearman rank coefficient was chosen as measure of correlation, because the relationships between scale measures and acoustics is not necessarily assumed to be linear.

### 3.3. Time domain evaluation

Traditional MOS ratings are usually computed over a whole segment. Since context is required for assessing specific qualities of a sample in question, these segments can not be made arbitrarily small. In this experiment we investigate whether participants are able to consistently mark parts of a given sample on a quality dimension. To this end, the subjects were presented with the same samples from 6 different systems and a digital representation of an oscillogram of the signal. The interface allowed them to mark regions by clicking and dragging the mouse. They were first tasked to provide an overall rating on the given dimension, with the scale items being integrated into one question. Then they were asked to mark the parts of the signal which they felt to be especially detrimental to the investigated dimension (e.g. very non-human or very emotionally negative). As in the previous experiments they were first presented with the same two anchors in a training phase to calibrate their internal expectations and thus reducing variability. To avoid forced choice artifacts they were also given the option to state that the system was equal in quality throughout the spoken parts on the dimension in question. We analysed the two major factors of "human-likeness" and "negative emotion" separately, with 10 participants each. The "audio quality" dimension was deemed to be unfit for this kind of examination, due to the fact that all its components describe variations of background artifacts which should be present throughout the signal.

## 4. Results

The factor analysis revealed 8 relevant underlying factors to the scales of quality in question. The cumulative explained variance amounts to 0.51 and Kaiser-Mayer-Olkin [38] $\mathrm{MSA} = 0.87$, with the lowest per-item values falling on the highly decorrelated measures of gender. The internal consistency of the scales was computed using Cronbach's Alpha [39] $\alpha = 0.90$ suggesting high correlation between the scales overall. Investigating item-to-total correlations to see which scales might actually be describing a separate construct yields very low scores for the *male* scale of $r = 0.06$. Additionally we find low bearings for the scales of *loud*: $r = -0.004$ and *native*: $r = -0.02$. Table 2 lists all items and their significant loadings ($> 0.4$) on the strongest correlating factor, with no single item having high complexity to significantly influence multiple factors. The factors are ordered by their explained variance of the overall data, with the most contributing factors appearing at the top.

### 4.1. Factors

The first factor describes the samples' "human-likeness" with the highest loadings being on artificiality and naturalness. Note that this factor conflates prosodic information such as "speech melody" with voice quality information like "metallic/tinny". This could point to the possibility that untrained participants are not able to differentiate between these constructs. The second factor labeled "audio quality" encompasses all items describing different variants of background artifacts. The third factor was named "negative emotion" as it seems to pertain to a combination of perceived voice qualities and subsequent elicited negative emotions in the listener. The fourth factor seems to describe terms which place the perceived speaker in an authoritative position, with the most influential loadings being confidence and authority. In factor five we find most of the scales associated with positive impressions and it is subsequently named "positive emotion". The sixth factor only contains the items of *calmness* and *agitatedness*. While this dimension has appeared in similar studies [13], two correlated items do not make for a salient and overdetermined factor and this dimension should as such be re-examined in confirmatory factor analysis. The seventh factor, labeled "seniority", seems to contain scales relating to the perceived speaker's age and voice quality. The least contributing factor is the orthogonal construct of gender which will be omitted in future investigations. Looking at the between factor interactions we find that the first two factors of "human-likeness" and "audio quality" show medium correlation with $\rho = 0.42$. We could not attest a strong correlation between the seemingly diametrically opposed factors of "positive emotion" and "negative emotion".

### 4.2. Consistency

The overall inter-rater consistency is fair with ICCs of 0.42, 0.40, 0.65 and 0.35 by sample, averaging to 0.45 across all audios and scales. Investigating the single scales, however, it quickly becomes apparent that this high overall consistency is mostly due to the items of *not male-male* and *not female-female*. These scales each obtained a Krippendorffs $\alpha = 0.98$ and $\alpha = 0.99$ respectively across samples. All other items under investigation were much more variant between participants' opinions, the closest being *non artificial-artificial* with $\alpha = 0.35$ and *non human-human* with $\alpha = 0.28$.

Table 2: *Synthetic quality scales and their strongest corresponding factor with the respective factor loading. Note that negative factor loadings denote inverse correlation and items with loadings < 0.4 have been omitted.*

| scale | label | loading |
|---|---|---|
| human | human-likeness | -0.69 |
| good speech melody | human-likeness | -0.43 |
| fluttering/pulsating | human-likeness | 0.43 |
| strange | human-likeness | 0.43 |
| irritating | human-likeness | 0.46 |
| metallic/tinny | human-likeness | 0.48 |
| interrupted/chopped | human-likeness | 0.61 |
| glitchy | human-likeness | 0.66 |
| artificial | human-likeness | 0.78 |
| unnatural/distorted | human-likeness | 0.82 |
| grainy | audio quality | 0.41 |
| hissing | audio quality | 0.60 |
| chirping/clicking | audio quality | 0.64 |
| rumbling | audio quality | 0.67 |
| crackling/static | audio quality | 0.79 |
| humming/buzzing | audio quality | 0.87 |
| frightening | negative emotion | 0.40 |
| quiet | negative emotion | 0.43 |
| dark | negative emotion | 0.58 |
| slow | negative emotion | 0.61 |
| sad | negative emotion | 0.66 |
| low | negative emotion | 0.68 |
| posh | dominance | 0.43 |
| loud | dominance | 0.47 |
| native | dominance | 0.50 |
| educated | dominance | 0.58 |
| stern | dominance | 0.58 |
| fluent | dominance | 0.59 |
| authoritative | dominance | 0.61 |
| confident | dominance | 0.78 |
| boring | positive emotion | -0.50 |
| emotive | positive emotion | 0.47 |
| captivating | positive emotion | 0.49 |
| pleasant | positive emotion | 0.52 |
| warm | positive emotion | 0.58 |
| calm | calmness | -0.66 |
| agitated | calmness | 0.51 |
| high | seniority | 0.47 |
| fast | seniority | 0.50 |
| thin | seniority | 0.58 |
| young | seniority | 0.62 |
| male | gender | -0.90 |
| female | gender | 0.94 |

### 4.3. Acoustic correlates

The results of the acoustic correlation analysis can be surveyed in tab. 3. All of the investigated acoustic measures strongly correlate with the perceived gender, with the strongest predictor being the fundamental frequency for all quality correlates. Outside of the gender factors the highest correlation could be attested between F0 and the high-low continuum as well as the not foreign-foreign scale. Also note that CPP seems to be largely independent of all perceptual quality dimensions under investigation, with the highest correlation also being gender at $\rho$(CPP, female)=0.23 and $\rho$(CPP, male)=$-0.22$. Within the acoustic measures we noted a strong correlation of jitter and shimmer to the fundamental frequency, which is to be expected

since the former are partially derived from the latter.

Table 3: *Spearman correlation between synthetic quality scales and common acoustic features. Only $\rho > |0.3|$ was included in this table.*

| acoustic feature | quality scale | $\rho$ |
|---|---|---|
| F0 | not female-female | 0.74 |
| jitter | not female-female | -0.50 |
| shimmer | not female-female | -0.72 |
| spectral slope 0-500Hz | not female-female | 0.64 |
| spectral slope 500-1500Hz | not female-female | -0.44 |
| Spectral Flux voiced | not female-female | -0.65 |
| Spetral Flux unvoiced | not female-female | -0.46 |
| F0 | not foreign-foreign | 0.33 |
| spectral slope 0-500Hz | not foreign-foreign | 0.32 |
| Spectral Flux voiced | not foreign-foreign | -0.33 |
| F0 | not high-high | 0.38 |
| jitter | not high-high | -0.31 |
| F0 | not low-low | -0.37 |
| spectral slope 0-500Hz | not low-low | -0.31 |
| F0 | not male-male | -0.74 |
| jitter | not male-male | 0.47 |
| shimmer | not male-male | 0.71 |
| spectral slope 0-500Hz | not male-male | -0.64 |
| spectral slope 500-1500Hz | not male-male | 0.50 |
| Spectral Flux voiced | not male-male | 0.66 |
| Spetral Flux unvoiced | not male-male | 0.48 |

### 4.4. Time domain analysis

Fig. 1 shows the participants' markings of all 6 samples on the human-likeness domain. As is evident, the whole marked amount of *unnaturalness* varies between systems. We also note that the participants vary in their individual granularity, with some participants marking whole chunks of the signal and others marking specific intervals. This leads us to believe that our instructions might not have been clear enough in asking participants to be highly specific in their selections. We report the inter-annotator agreement of participants within each domain with Fleiss kappa [40] where annotator agreement calculation is modified to consider pairs where one participant did not annotate any segments in the sample:

$$A_a = \frac{1}{\binom{K}{2}} \sum_{\ell=1}^{K-1} \sum_{m=\ell+1}^{K} \frac{\sum_{j=1}^{n} \text{mass}(j) \cdot \bar{\text{ov}}(j,\ell,m)}{\sum_{j=1}^{n} w(j,\ell,m) \cdot \text{mass}(j)}, \quad (2)$$

where $K$ is the number of participants, $\text{mass}(j)$ denotes the total length of marked segments in sample $j$ by any participant, $\bar{\text{ov}}(j,\ell,m)$ is the mean of relative overlap between marked segments of participants $\ell$ and $m$ over disjunct segments on sample $j$ and $w(j,\ell,m)$ is 1 if both participants $\ell$ and $m$ marked at least one segment in sample $j$, else 0. Tab. 4 shows the agreement depicted by system and condition. The human-likeness condition yields an overall moderate Fleiss kappa of 0.60. The value for the negative emotion domain is slightly lower with 0.55, suggesting that the signal properties of negative emotionality are not as clear. Analysing the agreement values by sample, we find that they also vary strongly between systems. Participants were also tasked to provide traditional ACR ratings on a five point scale for the domains in question, to serve as a comparative baseline for an inter-domain agreement measure. We computed a linear mixed effects regression with the

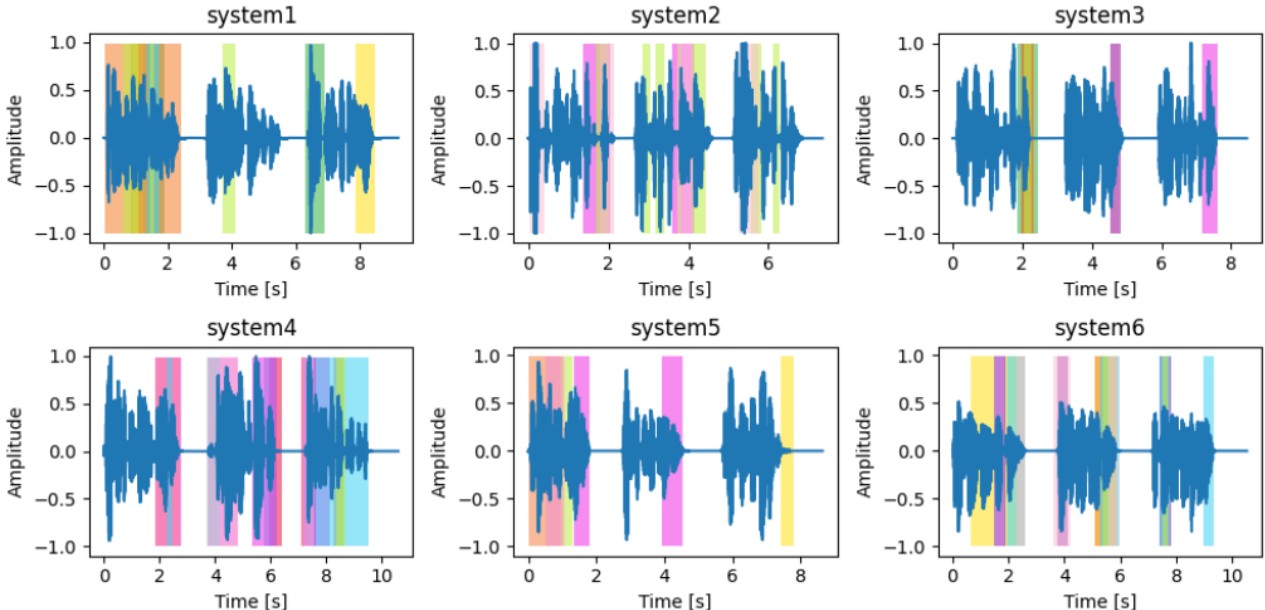

Figure 1: *Visualization of participants' markings of unnatural segments on 6 audio samples by different systems under the human-likeness condition. Each color denotes one participant, with overlapping segments showing multiple participants' agreement.*

audio samples as within factor and could not confirm an effect of the question being asked on participants ratings with $p > 0.5, \beta = -0.08$. To ensure that participants did not mark the same segments in both conditions we also compute a General Additive Mixed model (GAMM) on the summed participant markings over time. We model the density of participants markings at a given time step, dependent on question domain with the audio samples as within effect. The model finds a strong effect of the question domain on participants marked region density $p < 0.001, \beta = -0.67$. Fig. 2 shows the smoothed predictions of the computed model in both conditions. As is evident, the significant difference between the two sets of interval markings is in magnitude as was found by the intercept in our model, but barely in placement. This leads us to believe that despite the significant model participants did indeed mark similar regions within the audios independent of the dimension under investigation.

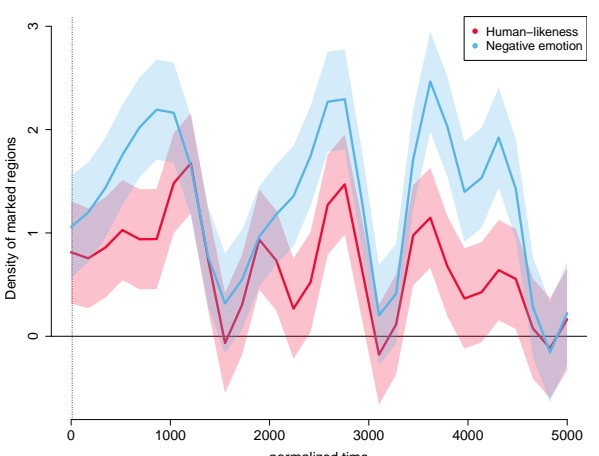

Figure 2: *Difference in density of participants markings of unnatural or emotionally negative regions, as predicted by a GAMM. Each curve represents the predicted amount of summed participant markings over the given time steps, with 95% confidence intervals. The time axis has been normalized across samples to allow for direct comparison.*

Table 4: *Kappa value, percentage of overlapping to total marked area and percentage of marked area to total signal length for time markings in the human-likeness and negative emotion domains.*

| | human-likeness | | negative emotion | |
|---|---|---|---|---|
| system: | % overlap | % of total | % overlap | % of total |
| system1 | 33.07 | 43.0 | / | 0.0 |
| system2 | 31.1 | 42.44 | 88.24 | 88.98 |
| system3 | 48.12 | 15.96 | 0.0 | 10.08 |
| system4 | 70.04 | 51.34 | 54.81 | 46.56 |
| system5 | 38.35 | 32.28 | 52.53 | 79.72 |
| system6 | 55.14 | 40.08 | 86.56 | 73.8 |
| $\kappa$ | 0.60 | | 0.55 | |

## 5. Discussion

The lack of inter-rater agreement on the same audio samples casts a shadow on the reliability of the common MOS procedures in speech synthesis evaluation. This runs counter to the previous findings on the reliability of MOS for signal degradation in [41], the consistency of listeners ratings in the blizzard challenge [8] or the retest reliability for naturalness MOS reported in [42], suggesting that the sample size for the reliability test might have been too small. In [43] it was shown, that bigger

sample sizes lead to more stable MOS values. A different explanation might be, that the level of abstraction of the terms queried is inversely correlated to the reliability of the results which would be in line with the findings in [42], who also reported utterance level correlations which were much lower than system level correlation to previous evaluations of the same data. This is also supported by the lack of effect we found on querying different domains between participants in the time marking experiment. While the validity of linear regression models on ordinal Likert type data is still an ongoing debate, other possible explanations could be the practice of merging subscales of a factor into one question as proxy items, or the fact that the domains were not presented at the same time to allow participants to rate them in the context of the whole construct. This second explanation could also be a reason for participants' time markings correlating across domains and could be remedied by having the subjects mark both domains simultaneously, which was decided against to reduce cognitive load. Regarding the poor correlation of acoustic measures to the perceived quality scales it should be noted that these findings do not suggest the acoustic measures are bad representations of their respective constructs. Rather, this points to the fact that these voice quality terms, which are ubiquitous in use for forensic and phonetic research, are not as well defined for a layperson and as such make for unstable quality measures in listening experiments. This interpretation has been corroborated in similar endeavours on finding acoustic correlates to the perceptual dimensions of voice [44], with [34] suggesting that there might not be one to one but compound relationships. Concerning the time domain evaluation procedure it should be noted that the subjects used in this pretest were recruited by word of mouth and 40% claimed to have semi-regular contact with synthetic voices. This might pose a confound regarding the findings of [45] that listeners do adapt to synthetic voices with exposure, albeit on intelligibility. On the other hand common challenge evaluation procedures recruit their participating scientists for listening evaluations and as such our subject base might be rather representative of standard evaluation conditions. Independent of the inter-rater agreement we also note that the time variant markings let us find patterns across participants data. Inspecting system3 in Fig. 1 for example, we clearly observe that the participants consistently marked the end of utterances, even though they did not agree on the same areas.

## 6. Conclusions

The construct of "synthetic speech quality" as a whole appears to be fairly stable on the dimensions of "naturalness" and "audio quality" as is evident when comparing the results to previous studies [29, 46] of similar nature. Our analysis did, however, uncover more dimensions in the "positive emotion", "negative emotion" and "dominance" categories. This is in line with the findings of [47] who found that affective scales have an effect on the overall perceived quality of experience in the context of personal digital assistants. The additional factors found in our study could be attributed to the larger set of initially administered scales, as well as the strict transformation to unilateral descriptive terms rather than qualitative questions. Regarding the examination of acoustic correlates it seems evident, even from our preliminary testings, that traditional acoustic measures do not serve as good representations for capturing participants' perceptual quality ratings on the investigated dimensions, as we could not confirm any monotonic relationship between the chosen measures and quality responses. First analyses of the newly proposed method to elicit participants' ratings

on a more fine grained scale yield promising results regarding the subjects consistency in marking the same regions. We do however also find strong overlap between the marked intervals of participants when being prompted to denote different aspects of quality, which warrants further investigation with a modified approach in which multiple factors are being queried at the same time. Following analysis of the marked regions with highest density across participants might also better serve to find acoustic correlates of the perceptual quality dimensions as well as yield insight into the individual shortcomings of the systems under investigation.

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
