# OpenReview forum: "Re-examining the quality dimensions of synthetic speech"
_Interspeech.org/2023/Workshop/SSW — SSW12_

### Official Review · Reviewer_74e8 · 2023-06-01
**This is an interesting paper that studies an importnat topic in speech synthesis research.**

**Rating:** 7
**Confidence:** 4

**Review:**

This paper studies the quality dimensions for evaluating synthetic speech. This is an important topic for speech synthesis research although few systematic studies has been conducted. This paper introduces the analysis procedures to deriving the 8 evaluation dimensions, and some acoustic analysis and time domain analysis have also been conducted. I have several comments.
1) There is no comparative analysis to show the deficiency of current widely-used evaluation dimensions, such as naturalness and similarity.
2) All contents for synthesis were from the Harvard dataset, which may not represent various application scenarios of speech synthesis.
3) Some descriptions need further clarification. For example, Section 3.2.1 mentions "15 distinct samples for each system", while the first paragraph in Section 3.2 says "18 per system". They are inconsistent and confusing.

---

### Official Review · Reviewer_r4KQ · 2023-06-01
**A thourough analysis of quality dimensions for synthetic speech**

**Rating:** 8
**Confidence:** 4

**Review:**

Key Strength of the paper:
The authors do a thorough analysis of additional dimensions for evaluating synthetic speech quality, besides the most common ones, like naturalness. The analysis is performed on a fine-grained level and has justified results.

Main Weakness of the paper:
Perhaps the large amount of factors and the fine-grained setup make it difficult to make absolutely confident assumptions, because the listener agreement in such a scale will vary.

Novelty/Originality, taking into account the relevance of the work for the SSW audience:
Novel work regarding synthetic speech quality, with proper justification of the results with multiple measures.

Technical Correctness:
Probably correct. In paragraph 3.1 it is mentioned that multiple terms were "freely supplied", which is a bit vague, but it is probably ammended later when it is mentioned that these terms were reduced with the help of experts.

Suggestions for improvement:
If possible a greater number of participants, as well as more samples being rated to verify that the overlaps and conclusions are consistent. Also, all systems used seem to be state-of-the-art, perhaps usage of some intended "bad" samples would verify some of the claims, as it will be checked if the listeners rate correctly the "intended bad aspect" of the generated audio.

Quality of References:
Adequate and good references.

Clarity of Presentation:
Very clear presentation.

---

### Decision · Program_Chairs · 2023-06-14

**Decision:**

Accept

**Comment:**

SSW2003 received 45 papers. The acceptance rate is 82%. We are pleased to inform you that your paper has been accepted by the SSW2023 Program Committee. Please read the reviews carefully and submit your camera-ready paper by June 28th. Most reviewers performed a detailed review. Please answer to their questions and consider their comments. Note that camera-ready papers are credited with one extra page to allow authors to consider reviewers’ suggestions. So max 7 pages in total including figures & refs.
The deadline for submitting the revised version (with full non-anonymized authors and refs!) is 28th June.